# Emergent Chess Skill Acquisition in Large Language Models

## Abstract

We investigate the emergent behaviours of rule comprehension, tactical execution, and strategic competence in transformer-based models trained on algebraic chess notation. To support structured reasoning, we introduce a disambiguation-aware tokenization scheme that explicitly encodes promotions, castling, checks, and mates, enabling fine-grained modeling of chess rules and dynamics. Our analysis reveals phase transitions in capabilities: shallow models fewer than 15 layers exhibit high illegality rates, while deeper models 20 layers or more increasingly demonstrate reliable tactical and positional behaviours. Training dynamics show while rule comprehension emerges early, higher-order abilities follow a hierarchical developmental path that mirrors curriculum learning. These trends remain consistent across decoding strategies and training distributions. Our findings suggest that transformer models can acquire human-aligned planning abilities in symbolic domains. Chess provides a tractable benchmark for evaluating the staged emergence of hierarchical competence in language models. Our methodology, including vocabulary design, architectural scaling, and behavioral evaluation, has the potential to generalize to other structured domains such as programming, formal logic, and mathematical proof systems.

## 1 Introduction

The rise of large-scale language models (LLMs) has raised fundamental questions about how these systems acquire structured decision-making abilities from data alone. While benchmarks often probe performance in static settings such as question answering or logical reasoning, far less is known about how sequence models internalize *dynamic, rule-governed domains* that demand planning, legality, and long-horizon consistency.

Chess provides an ideal setting for such study. It is bounded and interpretable, yet computationally vast: the number of legal board positions is estimated at roughly $10^{44}$ (Allis, 1994), while the space of possible distinct games, known as the Shannon number, exceeds $10^{120}$ (Shannon, 1950), dwarfing the number of atoms in the observable universe. With an average branching factor of 30–40 legal moves and typical games lasting around 80 plies, chess exhibits a game-tree complexity on the order of $10^{123}$ (Allis, 1994). This combination of strict rules and overwhelming combinatorial growth makes it uniquely suited for investigating how structured behaviours emerge in autoregressive models.

In this work, we study decoder-only transformers trained from scratch on human chess games in algebraic notation. Unlike prior work that emphasizes end-performance or reinforcement learning agents, our focus is on *training dynamics*: how rule comprehension, tactical motifs, and positional play emerge across model depth and training time.

Our approach is distinguished by a custom tokenization scheme that mirrors the syntax of chess notation, including explicit disambiguation tokens to handle ambiguous positions. This design enables models to generate contextually valid moves while respecting the rules of chess. We train across two datasets drawn from over a million human games with ELO $\geq$ 1600 and at least 40 moves: one filtered for high-quality white-win games and another balanced across outcomes. We compare models with 5, 10, 15, 20, and 25 layers.

Our analysis makes three contributions:

1. **Training Dynamics:** We provide the first systematic analysis of how rule comprehension, blunder avoidance, tactical motifs, and positional strategy emerge *as functions of both model depth and training time*. Prior work typically reports end-state performance; our focus is on the *developmental trajectory*.

2. **Strategic Complexity:** We move beyond legality and tactics to measure whether models learn simple strategies before complex ones, quantifying phase transitions in planning ability. This contrasts with earlier studies that treat gameplay competence as a monolithic outcome.

3. **Dataset Bias:** By comparing outcome-biased (white-win) vs. balanced datasets, we show how data distribution shapes model style—aggression, planning, and positional preference—providing insight into how training signals imprint strategic behaviours.

Together, these contributions recast chess not as an end in itself, but as a *microscope on emergent structure in sequence models*, offering insights relevant to symbolic reasoning and other rule-governed domains.

More broadly, chess serves here not as an end in itself but as a controlled testbed for studying how sequence models acquire structured, rule-governed behaviours. By treating chess as a microscope on emergent structure, our findings contribute to a broader understanding of how autoregressive training can give rise to rule compliance, abstraction, and strategy in complex environments.

## 2 RELATED WORK

Recent work has demonstrated that large language models (LLMs) can learn to play chess by training on textual representations of games, without explicit rule supervision or board state conditioning. Noever et al. (2020) showed that fine-tuning GPT-2 on PGN game data enables coherent move generation that respects opening principles, establishing the viability of autoregressive models in this domain.

Subsequent studies investigated whether such models internally track latent board states. Notably, Toshniwal et al. (2021) and Stöckl (2021) revealed that transformers trained solely on move sequences exhibit accurate legality prediction and internal piece tracking, even when perplexity remains flat. These works suggest that world modeling capabilities can emerge naturally from language modeling objectives.

Structured decoding and scaling have also improved play quality. Ruoss et al. (2024) showed that a 270M transformer can achieve ~2895 Elo without search, while Schultz et al. (2025) and Ye et al. (2025) leveraged LMs as policy evaluators or future-move samplers, improving planning through Monte Carlo Tree Search or diffusion rollouts.

In parallel, efforts like Feng et al. (2023) and Wang et al. (2025) paired move generation with strategy annotation, enriching model outputs with reasoning traces and achieving superior move quality. Zhang et al. (2025) emphasized the importance of uninterrupted long games during training, while Zhang et al. (2024) showed that models can exceed their training data's Elo through curated sampling.

Beyond the domain of chess, recent studies have turned toward understanding the evolution of skills and internal structure during pretraining. Bayazit et al. (2025) introduced sparse alignment methods to trace how specific linguistic features emerge and consolidate during LLM training. They demonstrate that core linguistic abstractions (e.g., syntax, irregular agreement) emerge in stages, and propose metrics like Relative Indirect Effects (RELIE) to quantify when specific features become causally important, offering a fine-grained view of conceptual acquisition over time.

Complementing this, Hakimi et al. (2025) used component-level analysis to track the functional roles of attention heads and feedforward networks throughout training, observing that models begin with general-purpose heads and later specialize, with some components being repurposed. They found that factual knowledge representations evolve hierarchically and remain plastic even in later stages, consequently supporting a dynamic view of neural specialization.

Our work bridges these threads by applying a curriculum-aware lens to chess modeling, treating model depth and training epochs as axes along which increasingly complex competencies emerge.

Unlike prior work focused on end-task performance, we analyze how rule-following, tactical execution, and strategic reasoning emerge over time, offering a behavioral analogue to recent mechanistic interpretability work. We introduce a structured tokenization scheme encoding algebraic disambiguation, castling, checks, and promotions, which enables interpretable tracking of skills like tactic execution, center control, and positional safety. Our findings highlight a layered trajectory of skill acquisition, marked by phase transitions in legality and an early stabilization of rule compliance, paralleling developmental patterns observed in both symbolic reasoning tasks and natural language pretraining.

# 3 DATA

## 3.1 DATA COLLECTION

We sourced training data from publicly available chess games on Lichess.org, a large-scale online platform with millions of user-submitted games. All games were downloaded in Portable Game Notation (PGN) format, which encodes move sequences in algebraic notation along with metadata such as Elo ratings, time controls, and outcomes. To ensure data quality, we retained only games in which both players had Elo ratings above 1600, filtering out noise from novice play while avoiding the idiosyncrasies of top-tier grandmasters. Games were further restricted to between 80 and 200 plies (40–100 full moves), excluding trivial early resignations and excessively long endgames. Finally, we constructed two datasets of 270,000 games for training and 23,000 games for validation to probe the effects of outcome distributions: a *white-win* dataset containing only White victories, and a *balanced* dataset with equal proportions of White wins, Black wins, and draws. Table 1 summarizes key style-related statistics across these datasets, showing broadly similar trends but with slightly higher tactical activity and material volatility in the White-win corpus.

Table 1: Statistics on playing styles by data type (mean $\pm$ std).

| Data type | Castling | Checks | Fork Rate | Pin Rate | Total Centipawn Loss |
|-----------|----------|--------|-----------|----------|----------------------|
| Balanced | $0.91 \pm 0.29$ | $2.51 \pm 2.49$ | $0.01 \pm 0.02$ | $0.02 \pm 0.03$ | $311.5 \pm 335.5$ |
| White | $0.85 \pm 0.36$ | $2.37 \pm 2.12$ | $0.02 \pm 0.02$ | $0.03 \pm 0.03$ | $414.6 \pm 404.0$ |

## 3.2 VOCABULARY DESIGN AND TOKENIZATION

To support fine-grained modeling of chess rules, tactics, and strategy, we developed a custom disambiguation-aware tokenization scheme based on algebraic notation. This design preserves game semantics and allows transformer models to learn directly from move sequences without auxiliary board supervision (e.g., FEN states). Moves with multiple legal origins are annotated with rank or file qualifiers (e.g., Nbd2, R1e1), while special moves such as castling (O-O, O-O-O) and promotions (e.g., e8=Q) are represented explicitly. Captures are consistently marked with x, including en passant, and suffixes denote checks (+) or checkmates (#). The resulting vocabulary encodes both target squares and semantically relevant features, maintaining move-order fidelity and supporting both subword and whole-token representations. This structured approach facilitates not only model training but also downstream interpretability, enabling analysis of learned behaviours such as tactic execution, castling patterns, and strategic development. Additionally as observed in A.5, while models trained on BPE tokenizer in similar experiments achieves lower overall illegality rates, models trained with the custom tokenizer exhibit more stable learning trajectories on complex pieces, suggesting that domain-specific structure supports smoother rule acquisition even when absolute performance lags. Vocabulary tokens can be further explored in A.6.

Table 2: Examples of our disambiguation-aware tokenization compared to standard algebraic notation (SAN).

| Move Type | Standard Algebraic Notation (SAN) | Tokenized Representation |
|---|---|---|
| Disambiguation | Nbd2 | `N DISAMBIG_FILE_b d2` |
| Disambiguation (capture) | R1xe4 | `R DISAMBIG_RANK_1 x e4` |
| Castling (kingside) | O-O | `O-O` |
| Castling (queenside) | O-O-O | `O-O-O` |
| Promotion | e8=Q | `e8 =Q` |
| Capture | exd5 | `e x d5` |
| Capture (en passant) | dxe6 | `d x e6` |
| Check | Qh5+ | `Q h5 +` |
| Checkmate | Qh7# | `Q h7 #` |

## 4 TRAINING IMPLEMENTATION

### 4.1 MODEL ARCHITECTURE AND LEARNING PARAMETERS

We adopt a decoder-only transformer architecture tailored for structured chess modeling, trained autoregressively to predict the next token in a sequence of algebraic moves. To isolate the effects of scale, we vary model depth across 5, 10, 15, 20, and 25 layers while holding other parameters fixed. Each model uses a hidden size of 768, eight attention heads, and a feedforward dimension of 1024, yielding between 20M and 100M parameters depending on depth. Positional information is encoded with rotary positional embeddings (RoPE), and we include special tokens for padding (`<PAD>`) and sequence termination (`<EOS>`).

Training is performed with causal cross-entropy loss, optimized using a cosine learning rate schedule with linear warmup. The base learning rate is set to $1 \times 10^{-5}$ with a warmup ratio of 0.1. We use a batch size of 8 and train for 10 epochs on approximately 270k games, which corresponds to roughly 33,000 batches per epoch. Input sequences consist of tokenized algebraic notation (e.g., `1. e4 1... c5 2. Nf3 ...`), and the model is optimized to predict the next token at each timestep. This token-level granularity enables fine-grained analysis of learning dynamics across both training epochs and architectural depth.

### 4.2 CHESS GAME SIMULATOR

To assess model gameplay in a dynamic and interactive context, we developed a simulator that alternates between model-generated and engine-generated moves until the game reaches a terminal state.

#### 4.2.1 SIMULATION PROCEDURE

Each simulation begins with the model playing as White. The game proceeds as follows:

1. **White move generation:** The model generates a move using temperature-controlled sampling. To enforce rule compliance, prefix-constrained decoding is applied via a legality trie built from the current board state.

2. **Prompt update:** The move is appended to the PGN-style prompt and applied to the board state using the internal simulator.

3. **Black move (Stockfish):** The Stockfish engine generates a reply using a fixed search depth or Elo cap. This move is parsed and appended to the prompt.

4. **Loop continuation:** The updated prompt (containing both White and Black moves) is passed to the model for the next White move.

This loop continues until the game reaches checkmate, stalemate, repetition, or another terminal condition.

### 4.2.2 Decoding and Legality Constraints

Our simulator supports several decoding strategies, including greedy decoding, top-$k$ sampling, top-$p$ (nucleus) sampling, and temperature-based sampling. These methods are used in comparative experiments to evaluate how different decoding regimes affect both gameplay quality and move legality.

To ensure that generated moves adhere to the rules of chess, we apply legality-constrained decoding through a dynamically updated trie of valid tokens derived from the current board state. The implementation differs slightly by context. For legality evaluation, logits are first generated over the full vocabulary, sampling is applied according to the chosen strategy, and candidate tokens are then validated against the legality trie; invalid moves are rejected and resampled. For full game simulation, logits are filtered in advance by masking all invalid tokens, and sampling is performed directly over this pruned distribution. In both cases, the legality trie is regenerated after each move, ensuring that only valid continuations remain possible. This approach guarantees coherent gameplay while preserving the intended diversity of decoding strategies.

## 5 Training Dynamics and Behavioral Metrics

In this section, we examine the progression of model performance and gameplay behavior across training epochs, model depths, and training data types. Our analysis spans a range of quantitative metrics, including loss curves, legality rates, tactical motif recognition, material evaluation, and positional strategy. Note that the legality metrics are computed over 10 simulated games, while the remaining three metrics are computed over 20 simulated games generated at each completed epoch during training.

Unless otherwise specified:

- **white** and **balanced** refer to the distribution of game outcomes in the training set; either exclusively white-win games or a balanced mix of white-win games, black-win games and draws.
- **nl** denotes the number of transformer layers and is used throughout figures to indicate model depth.
- All metrics correspond to model generated moves only.

All models are pretrained as described in Sections 3 and 4. Games were generated using a range of decoding strategies to assess not only predictive accuracy, but also the emergence of legal and strategic play. This section presents a detailed, metric-by-metric comparison of how capabilities such as rule comprehension, tactical reasoning, and strategic planning evolve with increased training time, architectural depth, and data composition.

### 5.1 Training and Validation Loss

Figure 1 shows training and validation loss over epochs, grouped by model depth and data type. All models exhibit smooth convergence, though shallower ones ($n = 5$, $n = 10$) plateau at higher loss, reflecting limited capacity. Deeper models ($n = 20$, $n = 25$) achieve lower final loss, especially on validation, indicating stronger generalization. Balanced-trained models slightly outperform white-only ones, though this advantage diminishes past $n = 15$. Overall, depth is the key driver of loss reduction, with data diversity playing a secondary role.

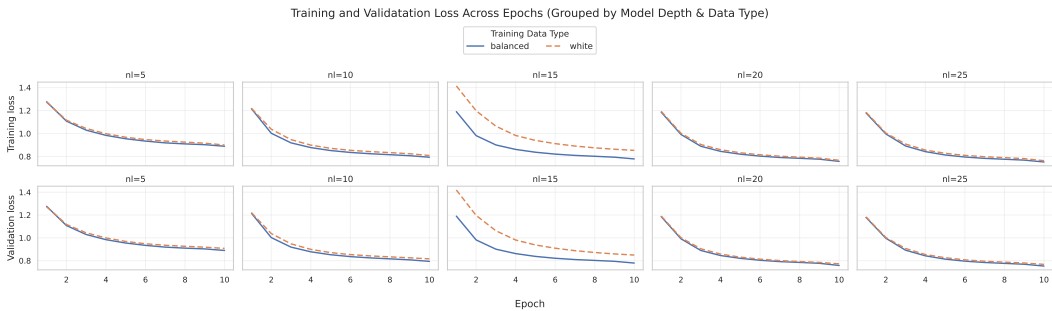

Figure 1: Training and validation loss across epochs, grouped by model depth (nl, number of layers) and training data type (balanced outcomes vs white wins only). Top row: training loss; bottom row: validation loss.

## 5.2 RULE COMPREHENSION

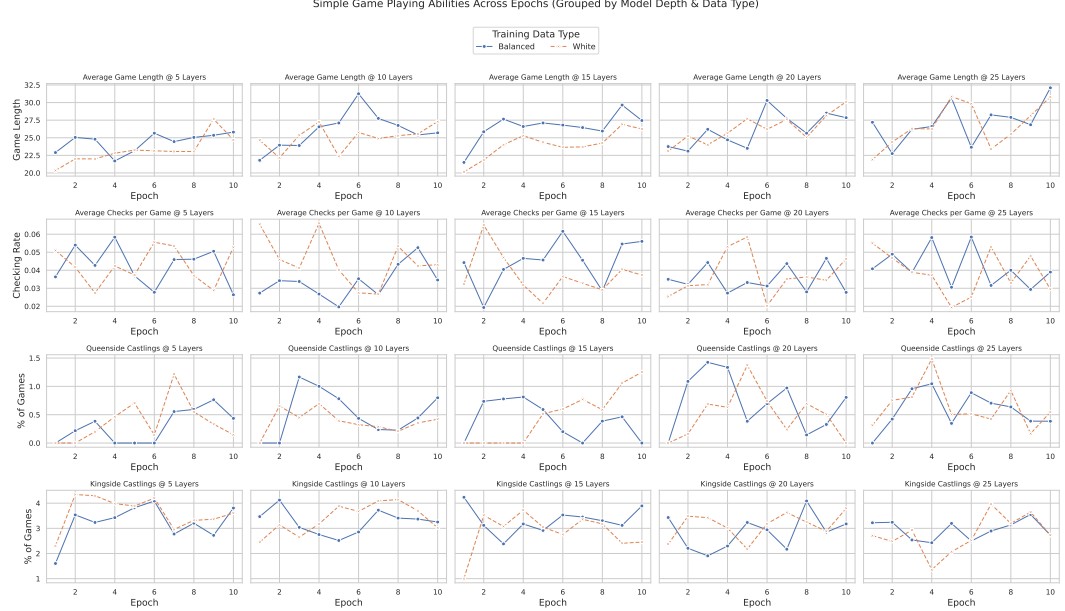

Figure 2: We report four metrics that reflect the progression of basic gameplay dynamics with training and model depth: (1) average game length, (2) average number of checks per game normalized by game length, and (3–4) percentage of games that include queenside or kingside castling, respectively.

To evaluate the emergence of rule comprehension, we generated 10 games per epoch and decoding strategy. As detailed in Appendix A.1 and illustrated in Figure 3, basic rule awareness emerges early in training, but legal move generation remains unreliable in shallower models ($n = 5$, $n = 10$), which exhibit persistently high illegality rates ($\sim$60–80%). A phase transition occurs around $n = 15$, where legality stabilizes and becomes a consistent behavior. Beyond this threshold, deeper models ($n = 20$, $n = 25$) generate predominantly legal moves ($\sim$10–20% illegality), indicating that sufficient architectural depth is essential for internalizing the game's rules.

The acquisition of individual piece movements follows a similar depth-dependent trajectory. Pawns are mastered earliest, with low illegality rates even from the first epoch. More complex pieces, namely knights, bishops, and queens, are learned reliably by epoch 4, but their accuracy remains sensitive to model depth, particularly for long-range movements or ambiguous positions. Full results, including decoding strategy and data type breakdowns, are presented in Appendix A.1.

While decoding strategy plays a secondary role, its influence is not negligible. Greedy decoding yields slightly higher legality, likely due to its deterministic bias toward high-probability tokens. Still, model depth is the dominant factor, suggesting that legality is primarily governed by internal representations rather than sampling behavior. Training data type also shapes performance: models trained on the balanced dataset consistently exhibit lower and more stable illegality rates than those trained on the white-only dataset. This advantage is especially pronounced for complex pieces such as rooks, bishops, and queens.

Figure 2 extends this analysis to broader game-play dynamics. Deeper models produce longer, more structured games, indicating the emergence of planning and restraint. Check frequency varies across depths, possibly reflecting competing objectives: giving checks vs. avoiding them. Castling behavior shows a clear developmental pattern: kingside castling is acquired earlier and more reliably; queenside castling is rarer and only appears consistently in deeper models. Once again, models trained on the balanced dataset show more stable castling trends, longer games, and richer dynamics overall. In contrast, white-trained models exhibit erratic castling behavior and noisier check distributions, suggesting more brittle or over-optimized playstyles. These results reinforce the role of training diversity in fostering more generalizable and procedurally complete rule comprehension.

## 5.3 MATERIAL LOSS

Material-based strategy can be studied in Figure 4 in Appendix A.2. Blunder rates decrease with both depth and training, showing that deeper models make fewer material-losing moves. Sacrifice recognition remains rare overall, but begins to emerge in deeper models, indicating that intentional material sacrifice is a late-acquired and more sophisticated capability. Good trade frequencies remain minimal, suggesting that evaluating and executing favorable exchanges is an advanced concept that requires additional training or depth to develop reliably.

Models trained on balanced and white-win games show varying performance. Blunder rates decline with training, with white-win models showing smoother and more stable convergence. Sacrifices in general remain rare but begin to appear in deeper balanced models, suggesting early signs of intentional material play. Good trades are virtually absent, with only sparse emergence beyond 15 layers, again favoring balanced models. Centipawn loss declines steadily across training, but differences between data types remain inconclusive with respect to whether either setting reflects more efficient or principled material management.

## 5.4 TACTICAL MOTIFS

In Appendix A.3, Figure 5 demonstrates the emergence of tactical pattern recognition across model depths. Fork recognition improves consistently with depth, reaching peak performance around $n = 20$–25. Pin recognition is more variable, suggesting it is a more challenging tactical concept to acquire. Skewers show depth-dependent variability, with some intermediate-depth models outperforming deeper ones. Discovered attacks remain relatively rare and stable across depths, indicating that this represents a late-emerging and advanced tactical capability.

Across most tactical motifs, models trained on the balanced dataset consistently exhibit higher motif rates, particularly at greater depths. In contrast, white-trained models show greater epoch-to-epoch volatility, with less reliable improvements in tactical behavior as depth increases.

## 5.5 POSITIONAL STRATEGY

The emergence of strategic behaviours across model depth and training is charted in Figure 6 in Appendix A.4. Opening development improves reliably with depth, indicating stronger coordination in early piece mobilization. King safety also increases, aligning with a decline in early-game defeats. By contrast, center control slightly fluctuates across depths and epochs, suggesting that this positional concept may require more nuanced modeling or reinforcement.

Middlegame metrics show more uneven trends. Space control and coordination improve gradually across both balanced and white-win models, while rook activity and outpost usage exhibit instability and remain underdeveloped, reflecting the greater challenge of encoding board-wide, multi-piece strategies.

These trends suggest a layered progression of competence. Foundational abilities, such as legality and piece development, stabilize between $n = 10$–15. Intermediate behaviours like castling and tactical motifs, emerge more reliably at $n = 15$–20. More advanced positional skills such as spatial dominance, rook activation, and coordination, only begin to surface at $n = 20$–25, and even then, often remain incomplete. The trajectory across training epochs mirrors this pattern: early epochs (2–4) establish basic rules, mid-epochs (4–8) introduce structure and tactics, and later epochs (8–10) refine behaviours, though sometimes at the cost of stability, possibly due to overfitting.

Dataset composition further shapes this development. Balanced-trained models show slightly smoother and stronger gains across most strategic metrics. In the opening phase, they outperform in development, king safety, and castling frequency, particularly at deeper layers. White-trained models tend to be noisier and less stable. In the middlegame, balanced models continue to lead in space control, rook activity, and outpost usage, while coordination is somewhat similar between the two models.

## 6 DISCUSSION

### 6.1 BASIC GAMEPLAY ABILITIES

The steady increase in game length with model depth suggests an emerging capacity for long-term planning. This is reinforced by reductions in centipawn loss[1] and blunders, indications of tactical soundness. While longer games could, in principle, result from indecision or repetition, here they correlate with improved positional control and fewer tactical collapses, pointing to meaningful gains in strategic coherence.

Strategic milestones such as checks and castling offer further insight. Check frequency follows non-monotonic trends across depths, suggesting that models are learning both to deliver checks and to avoid them, reflecting the development of offensive and defensive behavior. Castling tendencies reveal a clearer developmental trajectory: kingside castling emerges earlier and more reliably, while queenside castling appears only in deeper models. This asymmetry reflects hierarchical skill acquisition, where simpler strategies arise first, and more complex coordination (e.g., preparing queenside castling) depends on deeper representational capacity.

### 6.2 MATERIAL LOSS

Blunder rates decline steadily with model depth and training, suggesting that legality and short-term evaluation are internalized early. However, more sophisticated material reasoning such as sacrifices or favorable trades, remain rare and noisy. This asymmetry marks a developmental gap: legal competence and tactical avoidance emerge before deeper, value-based strategy. While the reduction in blunders reflects improved understanding of legal structure and tactical punishment, the sparse and inconsistent use of sacrifices and trades highlights limitations in the models' internal utility functions. Captures are not yet integrated into coherent long-term plans. Bridging this gap may require architectural changes, curriculum design, or auxiliary objectives that incentivize multi-step evaluation and counterfactual reasoning.

### 6.3 TACTICAL MOTIFS

Tactical motifs vary in difficulty. Forks are learned earliest, likely due to their local structure and high frequency. Pins and skewers require global board awareness and opponent modeling, and are acquired less consistently. Discovered attacks are especially rare, reflecting their reliance on deferred threat planning and latent piece alignment.

Motif usage improves with depth: shallow models may recognize isolated patterns, while deeper models begin to integrate them strategically. Nonetheless, even at 25 layers, performance remains inconsistent, suggesting that some motifs require abstract, multi-move inference beyond the capacity of sequence models trained on next-move prediction alone. This highlights potential for future work in curriculum learning, tactic-rich corpora, or auxiliary objectives targeting tactical abstraction.

---

[1] Centipawn loss quantifies the difference between the engine's evaluation of the move played and that of the optimal move, measured in hundredths of a pawn. Lower values indicate closer adherence to optimal play.

## 6.4 POSITIONAL STRATEGY

Limitations in material and tactical play are mirrored in the development of positional strategy. Foundational behaviours like development and king safety improve steadily with depth, whereas spatial concepts like space control, rook activity, and coordination lag behind. This suggests a need for broader board evaluation and long-range planning, which current next-token objectives may not fully support.

These patterns highlight a broader constraint: sequence models readily acquire legality and short-horizon heuristics but struggle with integrated, multi-phase strategic reasoning. Increasing model depth may help, but additional mechanisms may be needed to support deeper abstraction and utility tracking.

## 6.5 DATASET BIAS

The training distribution plays a central role in shaping procedural understanding. Models trained on balanced datasets consistently exhibit more robust gameplay: they generate longer games, castling more frequently and producing a wider variety of checks, demonstrating evidence of exposure to diverse strategic and tactical contexts. These conditions support generalizable rule learning, including rare mechanics like pawn promotion or castling constraints.

In contrast, white-win-only models often overfit to narrow, aggressive trajectories. They exhibit shorter games, limited castling, and lower strategic variability, suggesting a brittle reliance on frequent winning lines. This outcome bias impedes the acquisition of full-game procedures, particularly those requiring long-range planning or defensive play. Notably, performance gaps between the two regimes widen with depth: beyond $n = 15$, balanced-trained models consistently outperform. This reinforces a central insight that depth enables capacity, but data diversity enables competence.

## 7 CONCLUSION

Taken together, our results reveal a developmental trajectory in model gameplay. Early stages reflect syntactic competence, characterized by movement legality, blunder avoidance, and simple motifs. At greater depth, models begin to exhibit semantic competence through planning, positional structuring, and selective strategy. This progression from syntax to semantics, from local tactics to global planning, provides a framework for understanding how structured behaviours emerge in sequence models.

We also find that data diversity modulates this arc: models trained on balanced datasets exhibit smoother, more stable transitions between competence stages, while white-only models tend to develop brittle, aggressive heuristics that hinder semantic generalization. This suggests that representational capacity must be paired with training diversity to yield flexible, procedurally grounded play.

Persistent gaps in material valuation and advanced motif coordination underscore the limitations of next-move prediction as a sole learning signal, highlighting the need for additional supervision or architectural support to foster higher-level strategic reasoning. More broadly, these findings suggest that similar dynamics may govern learning in other rule-based domains, providing a template for studying the acquisition of structure and strategy in sequence models.

Future work could explore whether architectural modifications, curriculum learning, or targeted data augmentation can accelerate the emergence of strategic competence and unlock deeper abstraction in autoregressive models.

ACKNOWLEDGMENTS

The author acknowledges the use of AI tools for assistance with editing, phrasing, and refining functions related to the detection of chess behaviours. All research decisions and conclusions remain the sole responsibility of the authors.

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

# A   APPENDIX

## A.1   ADDITIONAL FIGURES - RULE COMPREHENSION

Figure 3: Acquisition of rule legality across training epochs and model depths. This figure illustrates the percentage of illegal moves generated over time, stratified by decoding strategy and model depth.

## A.2 Additional Figures - Material Loss

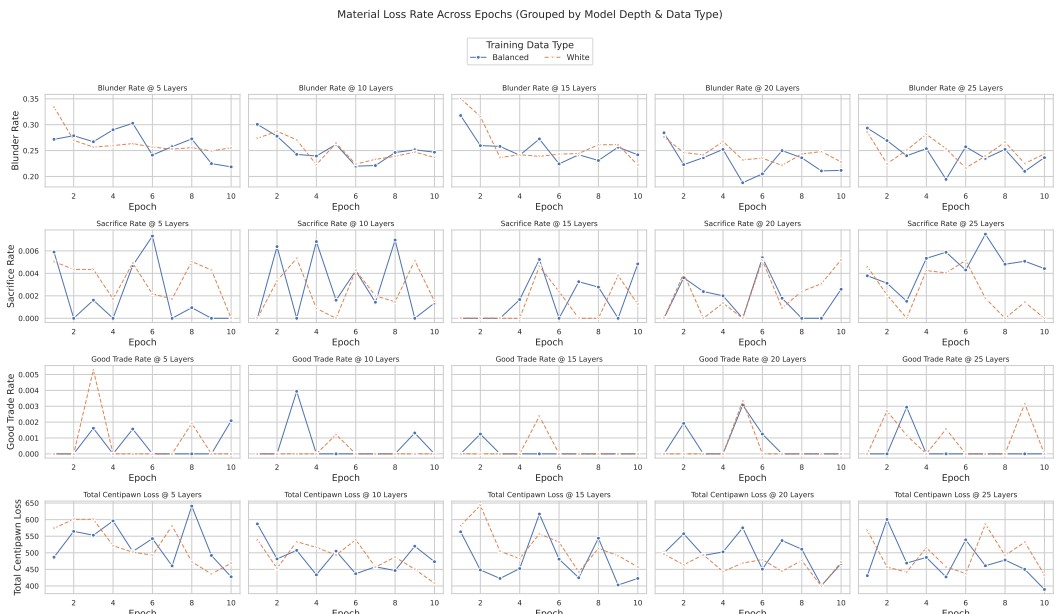

Figure 4: Emergence of material evaluation across model depth and training.

## A.3 Additional Figures - Tactical Motifs

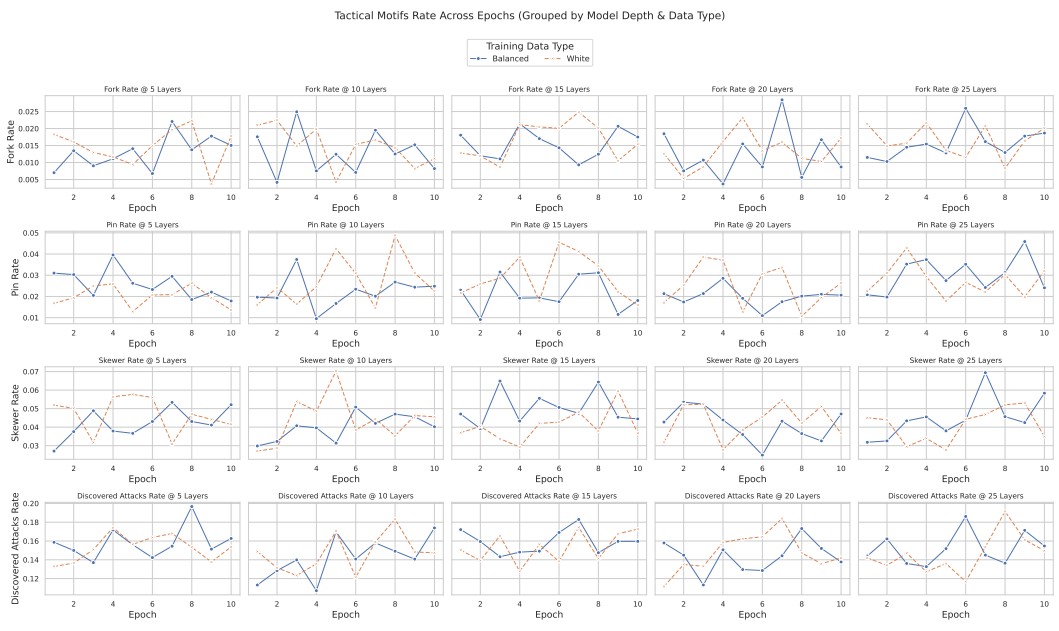

Figure 5: Emergence of tactical motifs across model depth and training. All tactics are normalized by game length, and the average rate is taken across all games.

## A.4  ADDITIONAL FIGURES - POSITIONAL STRATEGY

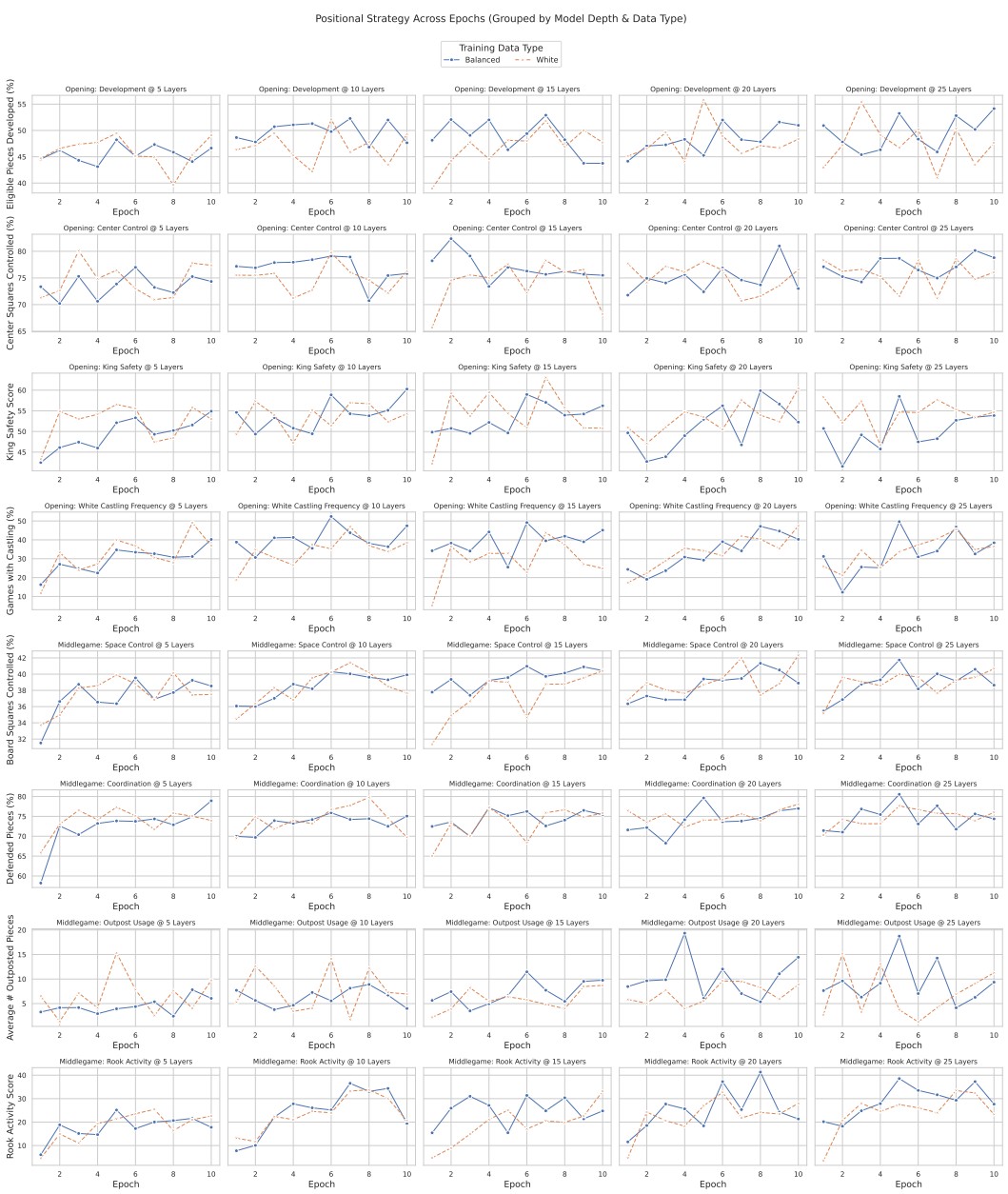

Figure 6: Positional strategy across epochs.

## A.5 Rule Comprehension Under BPE vs. Domain-Specific Tokenization

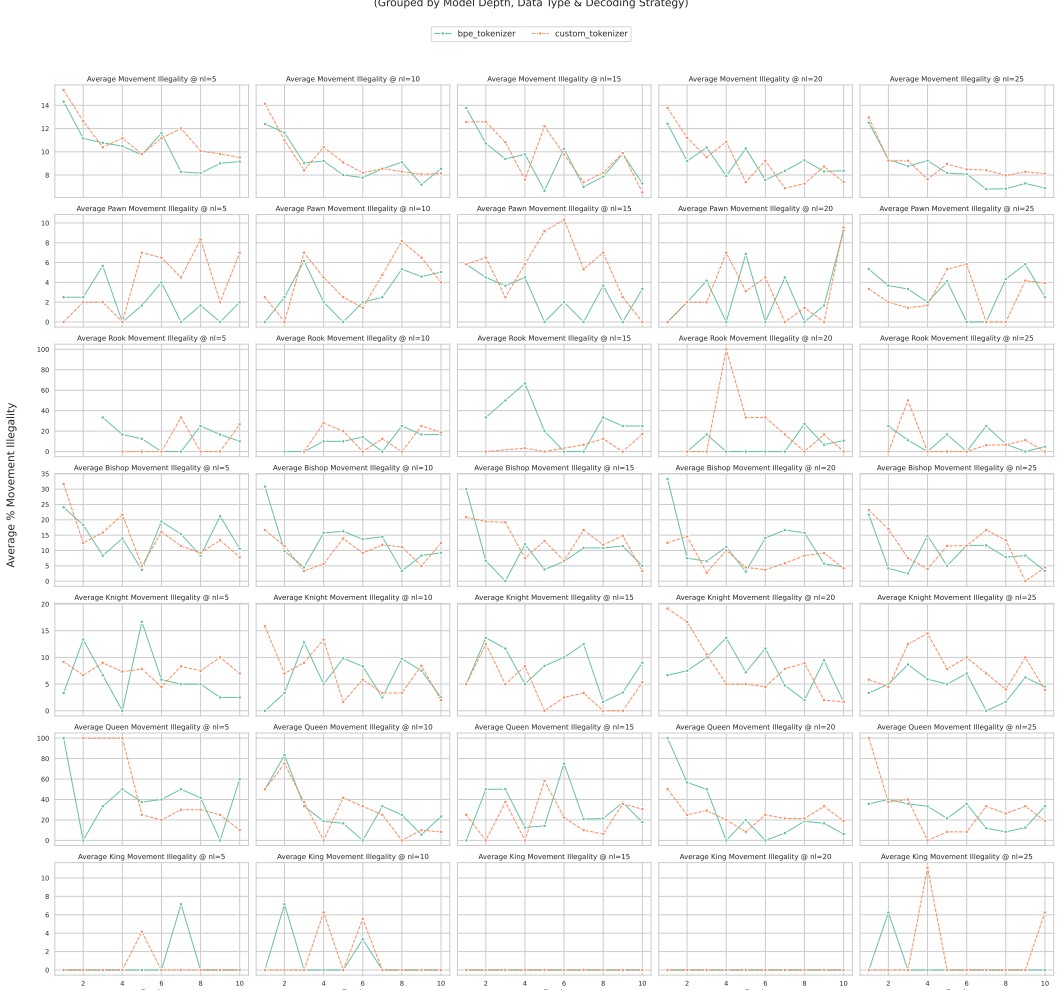

Figure 7: Average percentage of illegal model-generated moves across training epochs, grouped by model depth (columns), piece-specific movement type (rows), and decoding strategy. Curves compare models trained with a standard BPE tokenizer (green) against those trained with a chess-aware custom tokenizer (orange). Although BPE models generally show lower absolute illegality, the custom tokenizer demonstrates more stable learning patterns particularly on complex pieces, indicating that domain-specific tokenization can yield smoother rule acquisition despite higher baseline error rates.

## A.6 Vocabulary Tokens

### Move Markers

- #, +, . .

### Move Numbers

5

- 1., 2., 3., 4., 5., 6., 7., 8., 9., 10.,
- 11., 12., 13., 14., 15., 16., 17., 18., 19., 20.,

- 21.–298.

## SPECIAL TOKENS

- `<EOS>`

## PROMOTION SUFFIXES

- `=Q, =R, =B, =N`

## PIECE IDENTIFIERS

- `K, Q, R, B, N`

## CASTLING TOKENS

- `O-O, O-O-O`

## CAPTURE INDICATOR

- `x`

## FILES

- `a, b, c, d, e, f, g, h`

## SQUARES

4

- `a1--a8, b1--b8, c1--c8, d1--d8`
- `e1--e8, f1--f8, g1--g8, h1--h8`

## DISAMBIGUATION TOKENS

- `DISAMBIG_FILE_a--h`
- `DISAMBIG_RANK_1--8`

