# OpenReview forum: "Emergent Chess Skill Acquisition in Large Language Models"
_ICLR.cc/2026/Conference — Submitted to ICLR 2026_

### Official Review · Reviewer_35oD · 2025-10-28

**Soundness:** 3
**Presentation:** 2
**Contribution:** 2
**Rating:** 4
**Confidence:** 3

**Summary:**

This paper investigates chess skills in decoder-only transformer models trained from scratch on algebraic chess notation. The authors focus on the training dynamics and developmental trajectory of these skills, rather than final performance. They systematically vary model depth (5 to 25 layers) and the training data distribution (a balanced dataset vs. a white-win-only dataset). Using a custom, disambiguation-aware tokenization scheme, they analyze the emergence of three hierarchical levels of competence: rule comprehension, tactical execution, and strategic planning. The paper concludes that chess provides a valuable, interpretable benchmark for studying how structured, hierarchical reasoning emerges in language models.

**Strengths:**

The dynamics of skill acquisition rather than just end-state performance is interesting.

The study is well-designed varying variables: architectural depth and data distribution.

The evaluation is good, moving beyond simple win rates or Elo.

**Weaknesses:**

The current evaluation protocol appears to test the models as the White player. It would be beneficial to clarify if any experiments were conducted with the model playing as Black.

There seems to be a slight inconsistency in the evaluation methodology that I would appreciate clarification on. Rule comprehension is measured based on unconstrained generation, whereas the strategic evaluation uses prefix-constrained decoding to enforce legality. Could the authors explain the rationale for this dual approach? I wonder if this might decouple the model's strategic choices from its internal rule knowledge, potentially affecting the interpretation of the strategic metrics for shallower models that have not yet mastered legality.

The paper mentions that the training data was filtered to include games between 80 and 200 plies. Could the authors elaborate on the justification for this specific range?

The custom disambiguation-aware tokenization scheme is an interesting feature of the methodology. Could the authors explain why this hand-engineered approach was chosen over standard, data-driven subword tokenization methods like BPE?

**Questions:**

Please refer to the weaknesses

---

> ### Author Response · Authors · 2025-11-26
>
> Happy to clarify:
> - no experiments were conducted as Black.
>
> - For our evaluation methodology, there are two separate tasks; game simulation and rule comprehension, and we needed to isolate each task to evaluate the goal it served. When evaluating rule comprehension, we used 3 decoding strategies to study how well the model can capture the game's rules. When simulating games, legality was enforced in order to assess a model's performance and playing capabilities -- especially as this component would serve as the qualitative data needed to assess model strategy and tactics.
>
> - We chose to limit our dataset to chess games between 80 and 200 plies because the average game length between two experts is about 80 plies, and we capped at 200 since that was about the 75th percentile for game lengths therein. We didn't want games that were too short or too long when pretraining the model in the case of "troll" games or poor quality data, and decided early on that we can increase game length for other downstream tasks or more sophisticated curriculum training.
>
> - As for our tokenization scheme; tokens generated by BPE have sharp boundaries that may not translate to chess concepts. we wanted to design a minimal vocab that still contained core chess concepts that to retain inherent information about the domain within the vocab. Additionally, we'll be adding into our paper our experiments to compare BPE vs custom tokenization scheme which shows that although models trained with a BPE tokenizer had slightly higher rates of rule comprehension overall, models trained on our custom tokenization scheme had more stable learning patterns for more complex pieces like rooks and knights especially with increased depth.

---

### Official Review · Reviewer_nJCJ · 2025-11-01

**Soundness:** 2
**Presentation:** 3
**Contribution:** 2
**Rating:** 2
**Confidence:** 2

**Summary:**

Using chess as the research domain, the study examines how models acquire various chess skills from scratch. Lower-level skills, such as making legal moves, are learned early in training, whereas higher-level strategies, such as sacrificing pieces, are only acquired in the later stages.

**Strengths:**

Provides a detailed characterization of skill acquisition during the model’s training process.

**Weaknesses:**

1. **I am not an expert in explainable AI!**
2. I find the **article’s conclusion quite obvious: higher-level skills are learned later in training**. This is predictable and does not provide the reader with additional insights. I suggest the authors focus on discussing how the existing findings in the paper can inform better strategies for training models.

**Questions:**

see weakness

---

> ### Author Response · Authors · 2025-11-26
>
> Although that may be an intuitive conclusion, this is actually not as obvious - for example, there has been work such that suggest that order of capabilities learned are affected by training dynamics. See - https://arxiv.org/pdf/2406.19370, which suggests that learning order is really quite sophisticated. It is a pleasant observation in our paper that chess principles were able to be learned hierarchically.
>
> Additionally, this was an endeavour that was entirely domain specific -- the tokenization scheme was designed to have minimal vocab size while still conceptually retaining chess information. This is a valuable exercise in closed domains as it limits the vocab size as much as possible while still retaining inherent information about the domain within the vocab.

---

### Official Review · Reviewer_8rpj · 2025-11-01

**Soundness:** 2
**Presentation:** 3
**Contribution:** 2
**Rating:** 2
**Confidence:** 4

**Summary:**

The paper studies how language models acquire chess skills when trained on algebraic chess notation. By introducing a disambiguation-aware tokenization scheme and train models of varying depths (5-25 layers) on different datasets to study the emergence of capabilities. They observe clear developmental patterns: shallow models struggle with move legality, while deeper models develop tactical and positional understanding. Models trained on balanced game outcomes consistently outperform those trained only on white-win games.

**Strengths:**

- The paper is well-organized and clearly written.
- The intuition of this paper is great.

**Weaknesses:**

- The largest model studied (25 layers, ~100M parameters) is relatively small by current standards. It's unclear if the observed patterns would hold at scales of billions of parameters.
- The paper doesn't compare performance against purpose-built chess engines. This makes it difficult to assess overall performance compared to other methods.
- The paper lacks information about the computing resources needed for training.
- The paper lacks cast studies.

**Questions:**

Please refer to the "Weaknesses" section.

---

> ### Author Response · Authors · 2025-11-26
>
> 1. Although our models are relatively small, our hypothesis was methodically and correctly tested that scale does indeed impact learning ability even on such small configurations.
>
> 2. The models were trained using a T4 GPU on Google Colab .
>
> 3. The paper does contain several ablation studies for model configurations for legality and rule comprehension testing as well as complex game tactics and positional strategy, as well as a thorough investigation into related works. The former experiment for legality and rule comprehension also has multiple variations for decoding strategies as well. Is there anything else that we could add to bolster our case studies?

---

### Official Review · Reviewer_UMvn · 2025-11-02

**Soundness:** 2
**Presentation:** 3
**Contribution:** 2
**Rating:** 2
**Confidence:** 5

**Summary:**

This paper studies how language models acquire chess-playing abilities when trained on algebraic chess notation. The authors introduce a custom disambiguation-aware tokenization scheme and train models of varying depths on datasets. The paper reveals an approach similar to curriculum learning, with rule comprehension emerging early and higher-order abilities following later.

**Strengths:**

- The motivation of the paper is sound.
- The paper is well-structured with clear method descriptions and results presentation.

**Weaknesses:**

- The paper is titled with "Large Language Models." However, the maximum size of the models trained in the paper is 100M parameters, which is relatively small.
- As mentioned in Section 5.3, evaluations used only 10 games per configuration, which may limit the robustness of the proposed method, especially for cases like sacrifices or complex tactics.
- There's no analysis of how the custom tokenization scheme impacts learning compared to other alternatives.

**Questions:**

NA

---

> ### Author Response · Authors · 2025-11-26
>
> 1. Although what constitutes an LLM expands including its size, there is no widely accepted threshold that distinguishes a language model as "large", however we will update the title if that is a capability granted to us at this stage.
> 2. We are actively simulating more games, and results will follow shortly in order to more reliably measure complex tactics.
> 3. We have completed the analysis of BPE vs our custom tokenizer and will add the results to the appendix within the next day or so. We have found that although BPE has slightly better learning rates for rule comprehension overall, our custom tokenizer has more stable learning patterns for complex pieces like knights and rooks especially as model depth increases.

---

### Meta-Review · Area_Chair_RrRk · 2025-12-29

**Summary:**

The reviewers unanimously concluded that the manuscript is not currently suitable for publication.

**Reviewer Concerns:**

no concerns

**Reviewer Scores:**

not relevant

---

### Decision · Program_Chairs · 2026-01-26

Reject